Diversity structure of the microbial communities in the guts of four neotropical termite species

Vikram Surendra 1
Arneodo Joel D. 2
Calcagno Javier 3
Ortiz Maximiliano 1
Mon Maria Laura 2
Etcheverry Clara 4
http://orcid.org/0000-0001-8059-861X Cowan Don A. 1
Talia Paola 2 talia.paola@inta.gob.ar
1 Centre for Microbial Ecology and Genomics, Department of Biochemistry, Genetics and Microbiology, University of Pretoria , Pretoria, Gauteng , South Africa
2 Instituto de Agrobiotecnología y Biología Molecular (IABIMO), Instituto Nacional de Tecnología Agropecuaria (INTA), Consejo Nacional de Investigaciones Científicas y Técnicas (CONICET) , Hurlingham, Buenos Aires , Argentina
3 Centro de Ciencias Naturales, Ambientales y Antropológicas, Universidad Maimonides (CCNAA) , CABA , Argentina
4 Biología de los Invertebrados, Facultad de Ciencias Exactas y Naturales y Agrimensura, Universidad Nacional del Nordeste , Corrientes , Argentina
Souza Valeria
Electronic publication date: 2021 Apr 7
Publication date: 2021
Volume: 9
Electronic Location ID: e10959
Received 2020 Aug 24; Accepted 2021 Jan 27
Copyright: © 2021 Vikram et al.
Copyright year: 2021
Copyright holder: Vikram et al.
License: This is an open access article distributed under the terms of the Creative Commons Attribution License, which permits unrestricted use, distribution, reproduction and adaptation in any medium and for any purpose provided that it is properly attributed. For attribution, the original author(s), title, publication source (PeerJ) and either DOI or URL of the article must be cited.
License URL: https://creativecommons.org/licenses/by/4.0/

Keywords: Termite species, Gut microbiota, Prokaryotic and fungal diversity, Illumina amplicon sequencing

Funding: Fondo Argentino de Cooperación Internacional (FOAR) #6530 and #6745 Instituto Nacional de Tecnología Agropecuaria (INTA) PNAIyAV-1130034 Agencia Nacional de Promoción Científica y Tecnológica (ANPCyT) Proyectos de Investigación Científica y Tecnológica (PICT) 2018 No. 4149 This study was supported by grants from the Fondo Argentino de Cooperación Internacional—FOAR—(Ministerio de Relaciones Exteriores y Culto de Argentina) #6530 and #6745, the Instituto Nacional de Tecnología Agropecuaria (INTA) (PNAIyAV-1130034) and the Agencia Nacional de Promoción Científica y Tecnológica (ANPCyT) Proyectos de Investigación Científica y Tecnológica (PICT) 2018 No. 4149. The funders had no role in study design, data collection and analysis, decision to publish, or preparation of the manuscript.

==============================
The termite gut microbiome is dominated by lignocellulose degrading microorganisms. This study describes the intestinal microbiota of four Argentinian higher termite species with different feeding habits: Microcerotermes strunckii (hardwood), Nasutitermes corniger (softwood), Termes riograndensis (soil organic matter/grass) and Cornitermes cumulans (grass) by deep sequencing of amplified 16S rRNA and ITS genes. In addition, we have performed a taxonomic and gut community structure comparison incorporating into the analysis the previously reported microbiomes of additional termite species with varied diets. The bacterial phylum Spirochaetes was dominant in the guts of M. strunckii, N. corniger and C. cumulans, whereas Firmicutes predominated in the T. riograndensis gut microbiome. A single bacterial genus, Treponema (Spirochaetes), was dominant in all termite species, except for T. riograndensis. Both in our own sequenced samples and in the broader comparison, prokaryotic α-diversity was higher in the soil/grass feeders than in the wood feeders. Meanwhile, the β-diversity of prokaryotes and fungi was highly dissimilar among strict wood-feeders, whereas that of soil- and grass-feeders grouped more closely. Ascomycota and Basidiomycota were the only fungal phyla that could be identified in all gut samples, because of the lack of reference sequences in public databases. In summary, higher microbial diversity was recorded in termites with more versatile feeding sources, providing further evidence that diet, along with other factors (e.g., host taxonomy), influences the microbial community assembly in the termite gut.

Introduction

Termites are extremely efficient in degrading lignocellulose, and may be useful as “bioreactor models” for the conversion of lignocellulosic biomass into biofuels and other biomaterials (Brune, 2014).

Termites are broadly separated into “lower” and “higher” groups. The gut microbiota of lower termites consists of Bacteria, Archaea and Eucarya (such as flagellates and yeasts), whereas higher termites lack flagellated protozoans (Ni & Tokuda, 2013). These microbial symbionts have various roles in digestive processes, by participating in multiple functions including carbohydrate and nitrogen metabolism, oxygen and hydrogen consumption, N2 fixation, modifications of aromatic polymers and humification (Brune, 2014; Santana et al., 2015).

All termites feed on lignocellulose, the main component of plant cell walls. Lower termites (families Mastotermitidae, Kalotermitidae, Termopsidae, Hodotermitidae, Rhinotermitidae and Serritermitidae) have specific diets restricted to woody tissue, whereas higher termites (family Termitidae) have diverse feeding habits, which include wood, grass, fungi, lichen, litter, dung, humus and soil. Termitidae is the most diverse family of termites (around 75% of all species). This family comprises eight subfamilies: Apicotermitinae, Cubitermitinae, Foraminitermitinae, Macrotermitinae, Nasutitermitinae, Sphaerotermitinae, Syntermitinae and Termitinae. To date, 80 genera and 458 species, distributed in four subfamilies, Apicotermitinae, Nasutitermitinae, Syntermitinae and Termitinae, have been identified in the Neotropical Region (Krishna et al., 2013).

The termite gut microbiome was primarily believed to be determined by the host phylogeny, with influence from the diet (Hongoh, 2010; Rahman et al., 2015; Tai et al., 2015). More recently, Mikaelyan et al. (2015) suggested that the diet was the principal determinant of the higher termite gut microbiome composition showing that in all analysis of bacterial community structure, wood-feeding species were clearly separated from humus and soil feeders. Nevertheless, for each feeding source, a grouping of bacterial phylotypes by termite subfamily related to the taxonomy of the host was evidenced (Mikaelyan et al., 2015). Furthermore, Calusinska et al. (2020) investigated the adaptation of two higher termite colonies (Cortaritermes sp.) to Miscanthus (a perennial grass) consumption on laboratory conditions and constated the development of a diet-driven, adapted microbial consortium. Most authors agree that insect gut bacterial diversity is determined by environmental behavior, diet, developmental stage and host phylogeny (Hoback & Stanley, 2001; Yun et al., 2014; Rahman et al., 2015; Bourguignon et al., 2018; Dietrich et al., 2014). However, the relative influence of each of these factors is still not fully elucidated. In line with the above-mentioned studies, Rahman et al. (2015) concluded that even though the termite gut microbiome is mainly modulated by vertical inheritance, there may be adaptative changes in the microbial populations due to diet. Also, Dietrich et al. (2014) showed that phylogeny is not the unique factor influencing the termite microbiota composition, as they observed that changes in the diet or new niches can modify the bacterial community structure. Bourguignon et al. (2018) stated that termite gut microbiota is a result of a combination of vertical inheritance showing strong host specificity and horizontal transmission, where the latter can occur indirectly through the feeding substrates or via aggressive encounters. In addition, other authors conclude that there is a functional correlation between gut microbiomes from different termite hosts (Marynowska et al., 2020). They affirm that each termite species is a unique organism with its own gut microbiome and that there are functional similarities between microbial populations across different termite hosts.

In this study, we used prokaryotic 16S rRNA gene and fungal internal transcribed spacer (ITS) sequences to compare the gut microbiota of four higher termite species with different feeding habits and from three different subfamilies Cornitermes cumulans (Syntermitinae), Microcerotermes strunckii (Termitinae, Amitermes group), Nasutitermes corniger (Nasutermitinae) and Termes riograndensis (Termitinae, Termes group). Cornitermes cumulans builds mounds and feeds mainly on herbaceous material, predominantly grasses, but its diet may include cow dung and degraded wood (Souza et al., 2017). The soil-mound building termite T. riograndensis feeds mainly on soil and plant material. On the other hand, M. strunckii and N. corniger are strict wood-feeders, and therefore consume dry, wet or partially decayed hardwoods and softwoods (Scheffrahn et al., 2005). Here, we determined the structure of the bacterial and fungal communities in their guts and we performed a taxonomic and gut community structure comparison among these and other termite species previously reported. This is the first characterization of the gut microbiota of T. riograndensis and M. strunckii. Thus, this research provides novel information on the gut microbial communities of some unexplored termite species and contributes to shed light on the ecology and evolution of termites and their gut symbionts.

Materials and Methods

Insect collection

The termite species Cornitermes cumulans, Microcerotermes strunckii, Nasutitermes corniger and Termes riograndensis are widely distributed in Northeastern Argentina.

Specimens of C. cumulans (S 28°04′50.2″: W 58°16′12.1″) and T. riograndensis (S 27°25′29.7″: W 58°38′53.7″) were field-collected in Corrientes province, Argentina, from mounds located in grasslands consisting mainly of Andropogon lateralis Nees and Paspalum notatum Flüggé. M. strunckii (S 27°42′43.9″: W 59°13′35.2″) and N. corniger (S 27°27′38.3″: W 58°49′19.6″) were sampled in Chaco province from live trees Myracrodruon balansae (Engl.) Santin (hardwood) and Peltophorum dubium (Spreng.) Taub (softwood), respectively.

The termites were collected with the authorization of the Dirección de Recursos Naturales del Ministerio de Turismo de la provincia de Corrientes (permission number 845/13). No endangered or protected species were used in this study. The taxonomic identification of the termite species was inferred by morphology of the digestive tract of the workers caste specimens. Worker specimens were stored at −20 °C until further processing.

DNA extraction

Worker caste specimens were surface sterilized with 70% ethanol and their whole guts were dissected under a binocular microscope using sterile forceps. Ten dissected whole guts were pooled in a microtube containing RNA-later (Ambion, Grand Island, NE, USA); three independent extractions were performed per termite species. Microbial genomic DNA was extracted from the triplicate gut samples using the DNeasy Blood and Tissue kit (Qiagen, Frederick, MD, USA) according to the manufacturer’s instructions. In order to maximize the disruption of the gut tissues and their content, a thoroughly grinding with plastic pestles was performed prior to the chemical lysis.

The V4 hypervariable region of the bacterial and archaeal 16S rRNA gene was amplified using the specific barcoded primers 515F (5′-GTGCCAGCMGCCGCGGTAA-3′) (Turner et al., 1999) and 806R (5′-GGACTACNNGGGTATCTAAT-3′) (Caporaso et al., 2012). The ITS2 region of the ITS rDNA was amplified using the specific barcoded primers Forward (5′-GCATCGATGAAGAACGCAGC-3′) and Reverse (5′-ATATGTAGGATGAAGAACGYAGYRAA-3′) to assess fungal diversity. The samples were sequenced on an Illumina MiSeq instrument at the Molecular Research DNA (MR DNA) sequencing facility (Shallowater, TX, USA).

Bioinformatic and statistical analysis of 16S rRNA and ITS sequences

The paired end short reads were merged into single end reads and demultiplexed using the barcode sequences for each sample. The analyses of the 16S rRNA and ITS sequences were performed in the Qiime2 v2018.6 (https://qiime2.org) (Bolyen et al., 2019). Chimera identification and Amplicons Sequence Variants (ASVs) clustering were performed using DADA2 (Callahan et al., 2016) plugin in QIIME 2. The sequences were trimmed from the left at 35 base pairs (to remove leftover adapter and primer sequences) and truncated at 232 base pairs during the ASV clustering in DADA2. A broader community structure comparison was performed, including our data and those obtained previously from seven additional termite species by downloading the published database sequences (Bourguignon et al., 2018; Mikaelyan, Meuser & Brune, 2017) (Table 1). During the ASV clustering step sequences were trimmed 40 bases pairs from left (5′) (to remove any leftover adapter/primer sequence) and truncated at 200 base pairs during the DADA2 step. The sequences were assigned to ASVs using SILVA 128 16S rRNA (Quast et al., 2013) and UNITE (Abarenkov et al., 2010) database for Bacteria/Archaea and fungi, respectively.

Table 1 Summary of the higher termites used in the current study, their taxonomy classification, feeding groups and accession numbers of bacterial 16S rRNA amplicon libraries.

Host species	Subfamily	Diet preferences	Replicates	NCBI Biosample ID	References	
Nasutitermes corniger*	Nasutitermitinae	Wooda	(n = 3)	SAMN09635494	This work	
Nasutitermes sp.	Nasutitermitinae	Wooda	(n = 2)	SAMN08180495	Bourguignon et al. (2018)	
Microcerotermes sp. E	Termitinae	Wooda	(n = 1)	SAMN08180514	Bourguignon et al. (2018)	
Microcerotermes strunckii*	Termitinae	Wooda	(n = 3)	SAMN09635494	This work	
Microcerotermes parvus Mp193	Termitinae	Wooda	(n = 3)	SAMN04317068–SAMN04317070	Mikaelyan, Meuser & Brune (2017)	
Cornitermes cumulans*	Syntermitinae	Grassb	(n = 3)	SAMN09635494	This work	
Amitermes meridionalis	Termitinae	Grassc	(n = 1)	SAMN08180513	Bourguignon et al. (2018)	
Termes riograndensis*	Termitinae	Soil/Grass	(n = 3)	SAMN09635494	This work	
Neocapritermes taracua Nt197	Termitinae	Humusa	(n = 2)	SAMN04317074–SAMN04317076	Mikaelyan, Meuser & Brune (2017)	
Termes hospes Th196	Termitinae	Humusa	(n = 3)	SAMN04317083–SAMN04317085	Mikaelyan, Meuser & Brune (2017)	
Cornitermes sp. Co191	Syntermitinae	Littera	(n = 3)	SAMN04317065–SAMN04317067	Mikaelyan, Meuser & Brune (2017)	
Notes:

* Termites collected in this study.

a Based on the food types given for termite genera (Jones & Eggleton, 2011).

b Based on the dietary information in Souza et al. (2017).

c Based on dietary information in French & Ahmed (2011).

d Based on observations of Gontijo & Domingos (1991).

The resulting 16S rRNA representative sequences were aligned using MAFFT aligner (Katoh et al., 2002) and an unrooted tree was produced using FastTree 2 (Price, Dehal & Arkin, 2010). The tree was rooted at midpoint for the phylogenetic diversity analysis in QIIME v2. The α- and β-diversity indices for the 16S rRNA were analyzed in the QIIME 2 pipeline. In addition, α-diversity between the termite gut microbiomes was compared using the Kruskal-Wallis test, followed by false discovery rate (FDR) correction in Qiime2 pipeline. Weighted and unweighted UniFrac dissimilarities matrices were also obtained from the QIIME 2 pipeline (Lozupone et al., 2011). The Unifrac distances were plotted in phyloseq package (McMurdie & Holmes, 2013) using the Principal coordinate analysis (PCoA) (for four termite species from this work) and Non-metric multidimensional scaling (NMDS) (for 11 termite species comparison). The comparative analysis of 16S rRNA ASV table rarefied for 14,042 sequences in 10,736 ASVs (total number of sequences 393,176 for 27 samples). One sample corresponding to Neocapritermes taracua was removed from the further downstream analyses due to low number (8,210) of sequences obtained.

Alpha- and β-diversity indexes for the ITS sequences (fungal communities) were calculated and applied in the phyloseq (McMurdie & Holmes, 2013) and Vegan (Dixon, 2003) package in R. The test of significance for the β-diversity of ITS ASVs was performed after Hellinger transformation and Bray-Curtis distance matrix applied for the adonis function in Vegan package. The test of homogeneity of dispersion was performed using the betadisper function and Bray–Curtis dissimilarity in the Vegan package. The analysis of significant differences was performed based on the groups (termite diet (soil/grass, humus, litter and wood)) and host phylogeny (different subfamilies within Termitidae). A p value < 0.05 was set as the cutoff for significance of the statistical tests (The paired end short reads were merged into single end reads and demultiplexed using the barcode sequences for each sample). The analyses of the 16S rRNA were performed in the Qiime2 v2018.6 (https://qiime2.org) (Bolyen et al., 2019).

Accession numbers

The sequences obtained in this study are available in NCBI Sequence Read Archive (Bioproject PRJNA480379). The accession numbers are SRR7503210 (16S rRNA gene) and SRR7503211 (ITS).

Results

Illumina MiSeq sequencing of 16S rRNA gene and ITS amplicons, derived from gut samples from four termite species generated 1,761,565 high quality sequences. The 16S rRNA ASV table was rarefied to 79,256 sequences resulting in 4,882 ASVs (total number of sequences 951,072 for 12 samples). The ITS ASV table was rarefied to 7,418 sequences belonging to 149 ASVs (from a total of 66,762 sequences obtained from nine samples) (Table S1A). In addition, in the comparative analysis of the results from this study with previous works, a total number of sequences 393,176 for 27 samples were included; the 16S rRNA ASV table was rarefied for 14,042 sequences in 10,736 ASVs (Table S1B).

Bacterial, archaeal and fungal taxonomy

We identified 23 bacterial phyla in the termite guts of the four species tested. The dominant communities in the guts of wood-feeding termites (M. strunckii and N. corniger) were Spirochaetes (51% to 61%), followed by Fibrobacteres (~13%). Both species also showed similar relative abundances of Bacteroidetes (~8%) and Firmicutes (~8% in the case of N. corniger and slightly less, ~6%, for M. strunckii). Only a few reads (~2%) remained as unclassified bacteria (Fig. 1A). Thus, at the phylum level, gut communities of both wood-feeding species shared highly similar profiles regarding dominant taxa.

Figure 1 Relative abundance of bacteria, archaea (phyla level) and fungi (class level) in the gut of Neotropical termites.

(A) 16S rRNA gene (B) ITS sequence based taxonomic distribution in triplicate gut samples.

For the grass-feeding termite C. cumulans, Spirochaetes was again the dominant phylum (~44% of the total bacterial community), followed by Firmicutes (~23%) and Bacteroidetes (~13%). By contrast, the dominant phylum in the soil/grass-feeding species T. riograndensis was Firmicutes (~32%), followed by Spirochaetes (~20%) and Bacteroidetes (~13%). Only 2% of the sequences remained unclassified (Fig. 1A). Overall, Treponema was the most abundant bacterial genus in M. strunckii (59.1%), N. corniger (58.3%) and C. cumulans (42.8%). This genus was also found in T. riograndensis, though at a lower relative abundance (17.5%), where the genus Lactococcus predominated.

Of the Archaea, the only two phyla that could be identified, Euryarchaeota and Bathyarchaeota accounted for less than 2% of the reads, except for T. riograndensis, in which they represented almost 6% on average.

We also compared the gut bacterial and archaeal taxonomy and community structures found in this study with those of seven additional higher termites available in public databases. A total of 35 bacterial/archeal phyla were identified in the termite guts of the 11 species analyzed. Gut communities were dominated by Spirochaetes in the range of 10–74%, Firmicutes (4–63%), Bacteroidetes (0.2–70%) and Fibrobacteres (0.9–32%). Less abundant, though still well represented were Proteobacteria (<21%) and Actinobacteria (<19%), among others (Fig. S1). A little fraction of the reads could not be assigned to any phylum. As for the other phyla, the relative abundance of the Spirochaetes was variable within and between the diet groups. However, the highest abundance was observed in the wood-feeding termite group (up to 74%) followed by the soil- and/or grass-feeding group (up to 49%) and the litter-feeder termites Cornitermes sp. (consistently ~45%). Termites that feed on soil and/or grass, and humus showed high relative abundance of the phylum Firmicutes (up to ~55% for both groups, compared to ~15–25% for Cornitermes sp. and less than 14% for wood-feeders). One of the triplicates of the wood-feeder M. parvus exhibited an unexpectedly high proportion of Firmicutes, but this was not corroborated in any of the other two replicates. At the genus level most dominant taxa were Termite Treponema cluster followed by Treponema sp. regardless of the diet group (Fig. S2). For the ITS analysis, C. cumulans was excluded because of the low number of reads obtained. Regarding the other three termite species, the absence of matches with the available sequence data in UNITE (~81%, ~93% and ~99% of unclassified reads for N. corniger, T. riograndensis, and M. strunckii, respectively) did not allow the taxonomic placement of most of the fungal ASVs. The few taxa identified were assigned to the phyla Ascomycota and Basidiomycota, and within the former, the class Eurotiomycetes was the most abundant (Fig. 1B).

A rarefaction analysis performed for each gut sequence dataset retrieved rarefaction curves that reached a plateau for all samples, except for C. cumulans ITS sequences. This result suggests that the sample size was large enough to represent the bacterial and fungal diversity present in the communities (Fig. 2).

Figure 2 The rarefaction curve based on the species diversity showed sufficient coverage for the sequences.

The rarefaction analysis performed using the iNEXT package in R. The rarefaction curve based on the species diversity showed sufficient coverage for the sequences. The solid line (interpolated) is representing the actual sequence counts and the dashed line (extrapolated) is showing the predicted diversity. (A) Rarefaction curve of 11 termite species for the 16S rRNA sequences. (B) Rarefaction curve of 11 termite species for the 14,022 sequences (16S rRNA) in each sample. (C) Rarefaction curves of total ITS sequences for four termite species. (D) ITS sequences of the total ASVs and sequence counts showing enough coverage at the 7,418 sequences per sample. Termite species from this work are denoted by the red asterisk.

Diversity of prokaryotic and fungal taxa

The prokaryotic diversity of termite guts was analyzed using α- and β-diversity indices. The indices, Shannon, Pielou’s evenness and number of observed ASVs, showed no significant differences in α-diversity between the four newly reported termite microbiomes (Fig. 3). However, the α-diversity between diet groups (strict wood-feeders and soil/grass-feeders) significantly differed (Fig. S3). In addition, we evaluated the α-diversity between diet groups (soil/grass or strictly grass-feeders, humus-feeders, litter-feeders, and wood-feeders) incorporating sequence data from public databases (Fig. 4). Soil/Grass or strictly grass diet group showed highest number of observed ASVs, however, the comparison was only significantly different for the diet groups soil/grass-humus (Krukal–Wallis; H = 6.3, p = 0.011, q = 0.035) and soil/grass-wood (Kruskal–Wallis; H = 7.7, p = 0.005, q = 0.031) (Fig. 4C). The Pielou’s-evenness indices showed significant differences between the soil/grass-humus (H = 8.0, p = 0.004, q = 0.008) and soil/grass-wood (H = 9.2, p = 0.002, q = 0.007) groups. Also, a significant difference was observed between the humus-litter (H = 5.0, p = 0.025, q = 0.038) and humus-wood (H = 10.0, p = 0.001, q = 0.007) groups (Fig. 4B).

Figure 3 Prokaryotic α-diversity measures.

(A) Shannon, (B) Pielou’s evenness and (C) number of observed ASVs of the four species of termites. Comparisons were performed using the Kruskal–Wallis followed by Benjamini & Hochberg FDR correction.

Figure 4 Prokaryotic α-diversity measures.

(A) Shannon, (B) Pielou’s e venness and (C) number of observed ASVs of the total species of higher termites used to compare between diet groups. Comparisons were performed using the Kruskal–Wallis followed by Benjamini & Hochberg FDR correction. The H stats, p value and corrected p value (q) is written on the top of the paired comparison between the diet groups.

The prokaryotic β-diversity of the termite gut microbiome was compared using the unweighted UniFrac distances. The gut microbiome composition of the polyphagous soil/grass feeders was found to be similar and grouped distantly from that of the wood feeders, which were, in turn, separated from each other (Fig. 5A). The PERMANOVA test of Unifrac distances revealed that the β-diversity was significantly different for the four termite species, with a marked variation according to the diet group of termites (strict wood-feeders and soil/grass -feeders) (p < 0.05) (Table 2). Furthermore, the prokaryotic β-diversity of the new and database-retrieved termite gut microbiomes were compared using the weighted and unweighted UniFrac distances. Significant differences were observed on the unweighted UniFrac distances (p < 0.05), although no differences were found with weighted UniFrac distances (Table 2). Though, when considering diet groups, significant differences were found for both weighted (p = 0.005) and unweighted (p = 0.001) Unifrac distances. The NMDS of unweighted Unifrac distances revealed two separated clusters along the NMDS1 axis, the four gut microbiome studied in this work grouped distantly from those reported by Mikaelyan, Meuser & Brune (2017) who in turn were separated from those reported by Bourguignon et al. (2018) on the second axis (Fig. 5B). The NMDS of weighted Unifrac distances did not show grouping of the samples in the ordination plot based on the diet groups (Fig. 5C). The dissimilarity between the termite gut prokaryotic compositions for the diet group was compared using the betadisper (to test the homogeneity dispersion) and adonis (to test the similarity between the prokaryotic communities). The distance to group centroids based on Bray–Curtis distance were significantly different for diet groups (Betadisper, Permutest F = 19.607, p = 0.001). Adonis also showed significant differences (F(2.646) = 0.25, p = 0.001) in the community composition between the diet group of termites. However, when the termites were grouped according to the subfamilies they belong to, no significant differences were observed irrespective of the distance metric used (weighted or unweighted Unifrac) (Table 2).

Figure 5 Beta-diversity NMDS plot of Unifrac distances.

(A) PCoA plot of unweighted Unifrac distance. (B) NMDS plot of unweighted Unifrac distance of 11 termite species gut microbiomes (C) NMDS plot of weighted Unifrac distance of 11 termite species gut microbiomes. The green, red, yellow, brown colors are representing the grass, humus, litter and wood diet groups, respectively. The different shapes are representing the different termite species. Termite species from this work are denoted by the red asterisk.

Table 2 PERMANOVA analysis of bacteria/archaea and fungi.

16S rRNA		
	Four termite species (This work)	11 termite species	
	Unweighted Unifrac	Weighted Unifrac	Unweighted Unifrac	Weighted Unifrac	
Pseudo F	8.5445	67.0377	1.5075	1.5407	
p value	0.001	0.001	0.031	0.199	
	Diet groups (Grass and Wood)	Based on diet groups (Grass, Humus, Litter, and Wood)	
Pseudo F	4.5241	12.4488	3.74797	5.37108	
p value	0.002	0.002	0.001	0.005	
			Based on termite subfamily	
Pseudo F			1.61667	0.511453	
p value			0.097	0.568	
ITS (Bray-Curtis distance dissimilarity matrix)		
	Four termite species (This work)		
Pseudo F	13.1240		
p value	0.005		
	Diet groups (Grass and Wood)		
Pseudo F	3.8839		
p value	0.03		
Note:

The statistical analysis of ITS sequence data were performed based on the Hellinger transformation and Bray–Curtis distance-based dissimilarity matrix.

The analysis of the fungal community structure was restricted to our novel sequence data, since such information is still lacking in public databases. The Shannon and Pielou’s evenness α-diversity indices of fungal communities from M. strunckii, N. corniger and T. riograndensis gut samples differed significantly (Fig. S4). However, the number of observed ASVs showed no significant differences in α-diversity. Fungal α-diversity according to the diet group was also significantly different for the Shannon and Pielou’s evenness indices but not for the number of observed ASVs (Fig. S5).

To visualize overall similarities and differences in fungal community structure, we calculated Bray–Curtis distances between M. strunckii, N. corniger, and T. riograndensis, and displayed these analyses in the form of two-dimensional NMDS plots (Fig. S6). These analyses revealed that the fungal community composition in the gut samples of the three termite species was significantly different, whereas replicates of the same species were almost identical (Fig. S6). PERMANOVA analysis confirmed significant differences in the fungal communities of the different termite species and diet groups (Table 2).

Core microbiome

In total, 38 bacterial ASVs were shared across the four higher termite species reported here; which represent 28.5% of all the obtained sequences (Fig. 6A). The termite gut samples grouped according to their feeding habits shared 23 (wood feeders) and 38 (soil/grass feeders) additional ASVs. However, it has to be noted that N. corniger and T. riograndensis, which differ strongly in their feeding source and belong to distinct subfamilies, shared 31 additional ASVs (Fig. 6A).

Figure 6 Venn diagram showing the distribution of shared ASVs across the termite gut.

(A) Shared prokaryotic ASVs between four gut microbiomes, (B) shared prokaryotic ASVs between diet groups of termites, and (C) shared fungal ASVs between the three termites.

Of the 38 ASVs, 18 were assigned to Treponema sp., which represented 19% of the core microbiome sequences obtained (Fig. S6). These core ASVs were present in higher relative abundances in the guts of C. cumulans, M. strunckii and M. corniger than in the guts of T. riograndensis (Table S2; Fig. S7).

The analysis of core prokaryotic communities incorporating a larger number of host species (i.e., A. meridionalis, T. hospes, Cornitermes sp., Microcerotermes sp., M. parvus and Nasutitermes sp.) revealed no shared ASVs between the four diet groups of termites (Fig. 6B). However, the soil/grass-wood groups shared 188 ASVs. A high number of ASVs were unique to the termite species included in each diet group (soil/grass or grass: 3,729; humus: 2,046; liter: 1,285; wood: 3,379) (Fig. 6B).

We identified 11 fungal core ASVs, which represented an average of 71.3% of all the identified fungal sequences, shared in the gut samples of M. strunckii, N. corniger and T. riograndensis (Fig. 6C). However, the most abundant shared ASVs could not be assigned to any taxonomic level (Table S2).

Discussion

Numerous studies on the gut microbiota of higher termites have been published in the last two decades (Warnecke et al., 2007; Otani et al., 2014; Mikaelyan et al., 2015; Rahman et al., 2015; Santana et al., 2015; Su et al., 2016, Bourguignon et al., 2018). Knowing the factors that shape the structure of the microbial communities has become an increasing focus of interest. In this sense, relative few studies have dealt with the gut community composition in relation to the feeding habits of the host termite species.

This study provides a description of the intestinal microbiota associated with four Argentinian higher termite species (C. cumulans, M. strunckii, N. corniger and T. riograndensis), performed by high-throughput 16S rRNA gene and ITS amplicon sequencing analyses. Although data exist on the bacterial microbiota of C. cumulans and N. corniger (Dietrich et al., 2014; Köhler et al., 2012; Warnecke et al., 2007; Burnum et al., 2011; Mikaelyan, Meuser & Brune, 2017; Costa et al., 2013, Grieco et al., 2013, 2019); this is the first characterization of the gut microbiota of T. riograndensis and M. strunckii. Furthermore, very little is known about the fungal community in the microbiome of these and other termite species. These termites have different diet preferences ranging from hard- and softwood, to herbaceous materials and, soil/grass. Compared to softwood, hardwood harbors higher amounts of carbon content consisting on cellulose, hemicellulose and low proportion of lignin (Demirbas, 2005). Herbaceous plant materials have higher nutritious contents and lower lignin than wood, whereas soil contains diverse organic matter that is selectively utilized by the termite species. We explored the gut microbiota composition of the above-mentioned Argentinian higher termite species taking into consideration their different diet preferences.

Overall, Spirochaetes, followed by Fibrobacteres, Bacteroidetes and Firmicutes were the dominant gut phyla in termites feeding on wood or grass (M. strunckii, N. corniger and C. cumulans). By contrast, the dominant phylum in the soil/grass feeder T. riograndensis was Firmicutes, followed by Spirochaetes and Bacteroidetes. In accordance with our study, other researchers have reported Spirochaetes as one of the most abundant phyla in wood- and grass-feeding higher termite guts (Hongoh et al., 2005; Warnecke et al., 2007; Köhler et al., 2012; Brune, 2014; Dietrich et al., 2014; Mikaelyan et al., 2015; Rahman et al., 2015). In the last years the majority of glycosyl hydrolase genes encoding putative cellulases and hemicellulases, identified by metagenomic and metatranscriptomic studies, have been associated with Spirochaetes, Fibrobacteres, Bacteroidetes and Firmicutes (Warnecke et al., 2007; He et al., 2013; Ben Guerrero et al., 2015; Grieco et al., 2019; Calusinska et al., 2020; Marynowska et al., 2020; Romero Victorica et al., 2020).

The high abundance of Spirochaetes (including the genus Treponema sp.), Fibrobacteres and Bacteroidetes in wood-feeding termites may be related to the nitrogen fixation and lignocellulosic processes (Lilburn et al., 2001; Breznak, 2002; Warnecke et al., 2007; Yamada et al., 2007; Su et al., 2016). In addition, high abundance of Firmicutes (mostly Ruminococcacae) in T. riograndensis was in accordance with previous reports of soil- and humus-feeding termites (He et al., 2013; Dietrich et al., 2014; Mikaelyan et al., 2015; Santana et al., 2015).

In particular, several studies have reported a proportion of Spirochaetes of approximately 50–60% of total prokaryotic population in the gut microbiome of N. corniger, similar to that reported here (Warnecke et al., 2007; He et al., 2013; Dietrich et al., 2014; Santana et al., 2015; Su et al., 2016). However, Köhler et al. (2012) observed a lower proportion of Spirochaetes in N. corniger and N. takasagoensis. This discrepancy could be due to variations in DNA extraction methods (Morgan, Darling & Eisen, 2010) and/or the use of different PCR oligonucleotides (Engelbrektson et al., 2010).

A low proportion of Archaea was present in the gut community profiles of the four higher termites gut communities analyzed. The detected archaeal phyla were Euryarchaeota and Bathyarchaeota. Euryarchaeota includes closely related genera already known for their methanogenic activity (Rahman et al., 2015). The methanogenic archaeon Methanimicrococcus sp. was present in the core microbiome albeit in different proportions. This genus had already been detected in the gut microbiomes of other higher termites and cockroaches (Paul et al., 2012).

The extremely high relative abundance of Treponema spp. (phylum Spirochaetes) in the core of C. cumulans, M. strunckii and N. corniger, and at a lower extent in that of T. riograndensis, suggests that Treponema genus has an important role on the overall physiology and digestive processes of wood- and grass feeding higher termites. The predominance of Treponema in the termite gut microbiota has been pointed out by several authors (Warnecke et al., 2007; Köhler et al., 2012; Shi et al., 2013; Benjamino & Graf, 2016). Microbiome diversity is thought to be mainly related to the phylogeny of the termite and also, to their diet habits. In order to infer possible relationships between gut microbial community, phylogeny, and diet; the gut communities of additional termite species reported elsewhere have been included for a broader comparison. Again, Spirochaetes, Firmicutes, Bacteroidetes and Fibrobacteres were the dominant phyla. Also the most dominant taxa were Termite Treponema cluster followed by Treponema sp.

In the four species evaluated in this study, the diversity of the fungal community was markedly lower than that of prokaryotes. The high proportion of taxonomically unclassified fungal ASVs may result from the lack of representative sequences in the UNITE database (Hongoh, 2010; Santana et al., 2015). Although fungi are not as prevalent as bacteria in higher termite guts, an important unresolved issue is to determine the role of fungi in cellulolytic processes development and fitness. Some of these functions could be to provide a nitrogen source, degrade high molecular weight molecules and produce pheromones for mating and communication (De León et al., 2016; Zhang et al., 2018). However, the role of fungal microbiota in these processes is not clear yet (Brune, 2014).

The fungi classes that could be identified in the core were Eurotiomycetes and Malasseziomycetes. Eurotiomycetidae are producers of secondary metabolites, fermentation agents and xerophile and psychrophile enzymes. They had been previously reported in the gut of the litter-feeding termite Synthermes wheeleri (Santana et al., 2015). Malasseziomycetes are ecologically diverse and wide spread yeasts. The genus Malassezia includes lipophilic yeasts and has been known as a common inhabitant of human skin (Paulino, Tseng & Blaser, 2008). A report by Zhang, Su & Blackwell (2003) also identified this yeast in the guts of beetles.

The presence of a common core microbiota suggests that these taxa are retained despite differences in habitat, geography and food source, and regardless of host phylogeny. This core composition may be important for the maintenance key functions and may serve as the basis for microbial community resistance and/or resilience (Huse et al., 2012; Shade & Handelsman, 2012; Benjamino & Graf, 2016). However, when sequences obtained from more termite species were added to the analysis, no shared ASVs between all four diet groups of termites was evident, suggesting that both host termite phylogeny and diet (and eventually other additional factors) can influence the community structures of gut microbiota.

The α-diversity of gut bacterial communities in the soil/grass feeder group was significantly higher than that in wood feeders. Among the former group, the gut microbiota diversity in T. riograndensis (a soil- and grass-feeding termites) was higher than in species that fed on grass only (C. cumulans). The lower microbial diversity found in the wood-feeding termites may be related to the maintenance of a more specialized microbiota that is necessary for performing an efficient lignocellulose metabolism, and therefore for the host survival (Breznak & Brune, 1994; Colman, Toolson & Takacs-Vesbach, 2012). The studied species of wood-feeding termites have a very limited diet, which includes complex carbohydrates (cellulose, hemicellulose and lignin), and this characteristic may explain the lower α-diversity. Feeding on live trees may expose the host and potentially its microbiota to tree physiological responses (Morewood et al., 2004), which may further shape gut community dynamics. The higher α-diversity in the soil/grass feeding termites could be related to the diverse range of carbon and nitrogen sources available in their diets; as more complex substrates require more complex degradative capacity and therefore more complex communities. On the other hand, the host habitat also may influence the relative bacterial abundances of the termite gut microbiota (Yun et al., 2014). When including in the analysis previously reported sequences obtained from additional termite species, the bacterial α-diversity indices (Shannon and Pielou’s-evenness) and number of observed ASVs found in the soil/grass or strictly grass diet group were, again, significantly higher than in wood-feeders. Significant differences were also evidenced for some indices between the other feeding groups suggesting that multiple factors, which may include diet and host taxonomy, contribute to shape the gut microbiota.

The intercommunity analysis restricted to our four Argentinian termite species showed that the gut microbiomes of soil/grass feeders were clearly separated from those of wood feeders. Soil/grass-feeding termite species grouped closely, whereas the wood-feeders were spatially separated from each other. The replicates of C. cumulans, T. riograndensis and M. strunckii showed little variation, whereas those of the wood feeder N. corniger were more disperse. The termites M. strunckii and N. corniger were sampled from alive trees of Myracrodruon balansae (hardwood) and Peltophorum dubium (softwood). Even though the relative abundance at the phylum level was similar, microbial species composition was different between both termite species.

The β-diversity analysis including previously reported gut microbiomes showed that the differences between communities were due to the presence of distinct ASVs rather than to changes in relative abundance. A permutation-based test on the unweighted Unifrac distances showed significant differences between diets, although no differences were found with weighted Unifrac distances. The same analysis applied to diet groups showed significant differences among them. However, non-significant clustering of the termites regarding host taxonomy (subfamilies) was observed. Although the unweighted Unifrac distances revealed two separated clusters along the NMDS1 axis, these groupings are not related to diet or host phylogeny.

Test of homogeneity dispersion using betadisper suggested that the sample groups are having significantly low variance in the community dispersal (p < 0.001). PERMANOVA test using the adonis function also suggested the significant differences in the termite gut community using the diet as a grouping factor. However, non-significant differences were observed grouping by host phylogeny. The inter-host comparison among microbial communities based on metagenomic data obtained by different research groups should be approached carefully. Besides methodological heterogeneity concerning microbial DNA extraction, amplification and analysis (Morgan, Darling & Eisen, 2010; Engelbrektson et al., 2010), several other issues, starting at insect collection, are also to be considered. The specimen’s provenance (e.g., substrate origin, single or multiple colonies sampled, etc.), the number of pool replicates, to mention a few items, may influence the results substantially. For instance, during a prospection of culturable Cohnella-related bacteria in the guts of three Neotropical termites, the sequenced clones tended to group according to their host species and to the different colonies from which insects were sampled, too (Arneodo et al., 2019). Thus, even though much data are available, a multiplicity of environmental, methodological and technical variables makes a broad comparative analysis difficult (Pollock et al., 2018). Some apparent inconsistencies between authors appear, and analyses (especially those regarding diversity) may reflect not only the intrinsic characteristics of the studied microbiotas but also the different experimental conditions.

The understanding of the termite-microbiome interaction requires the exploration of the composition and structure of the microbiota, as well as the characterization of its main metabolic activities in different taxonomic termite groups with different types of diets. Altogether, and concerning the four Argentinian termite species, no obvious pattern was observed in the microbial community structures, except for the similar relative abundance of bacterial phyla in the case of strict wood-feeders. This provides further evidence that the gut microbiota composition is the result of multiple factors, which may include (but not be limited to) diet and host taxonomy.

Conclusions

We have explored prokaryotic and fungal community structures in the guts of four higher termite species collected in NE Argentina: Nasutitermes corniger, Cornitermes cumulans, Termes riograndensis and Microcerotermes strunkii. For the latter two species, this study constitutes the first characterization of their associated microbiota. Also, we have performed a taxonomic and gut community structure comparison incorporating into the analysis with previously published microbiome data sets of termites with different diet preferences. The bacterial phylum Spirochaetes (in particular, the genus Treponema sp.), was dominant in the guts of M. strunckii, N. corniger and C. cumulans, whereas Firmicutes predominated in the T. riograndensis gut microbiome. In the broader analysis, also Spirochetes was the dominant phylum followed by Firmicutes, Bacteroidetes and Fibrobacteres. Both in our own sequenced samples and in the extensive comparison, prokaryotic α-diversity was higher in the soil/grass feeders than in the wood feeders. In addition, the β-diversity of prokaryotes and fungi was highly dissimilar among strict wood-feeders, whereas that of soil- and grass-feeders grouped more closely. However in the broad comparison the β-diversity analysis showed significant differences regarding diets but non-significant clustering of the termites for the host phylogeny groups. Concerning fungi, our work provides new insights on a poorly studied field. Ascomycota and Basidiomycota were the only fungal phyla that could be identified in all gut samples, because of the lack of reference sequences in public databases. This study provides evidence that communities are shaped by multiple factors that may include, among others, diet and host taxonomy.

Supplemental Information

Supplemental Information 1 Taxonomy plot for the termite samples grouped into diet groups. The plot is showing the relative abundance of the prokaryotic phyla in the gut of termites.

Click here for additional data file.

Supplemental Information 2 Taxonomy stack bar-plot of the prokaryotic communities in the diet group. (A). Phylum level taxonomic classification (B) Genus level taxonomic classification. Taxa less than 2% were combined and named as a “<2% abund.”.

(A). Phylum level taxonomic classification (B) Genus level taxonomic classification. Taxa less than 2% were combined and named as a “<2% abund.”.

Click here for additional data file.

Supplemental Information 3 Prokaryotic α-diversity measures (Shannon, Pieolou’s evenness and number of observed ASVs) according to diet groups (strict wood-feeders versus soil/grass-feeders).

Comparisons were performed using the Kruskal-Wallis followed by FDR method in Qiime2.

Click here for additional data file.

Supplemental Information 4 Fungal α-diversity measures (Shannon, number of observed ASVs and Pieolou’s evenness) in three species of termites.

Comparisons were performed using Kruskal-Wallis followed by FDR method in Qiime2.

Click here for additional data file.

Supplemental Information 5 Fungal α-diversity measures (Shannon, number of observed ASVs and Pieolou’s evenness) according to diet groups (strict wood-feeders versus soil/grass-feeders).

Comparisons were performed using Kruskal-Wallis followed by FDR method in Qiime2.

Click here for additional data file.

Supplemental Information 6 NMDS plot for the three termite gut samples using Bray Curtis analysis for ITS sequences.

Click here for additional data file.

Supplemental Information 7 Relative abundance of 38 core ASVs at classified taxa level in the gut microbiome of C. cumulans, M. strunckii, N. corniger and T. riograndensis. Redder and lighter red indicate greater and less abundance, respectively.

Click here for additional data file.

Supplemental Information 8 Number of input and final number of reads for the 16S rRNA and ITS sequence data (S1A). Number of input and output 16S rRNA sequences obtained in the comparative analysis (S1B).

Click here for additional data file.

Supplemental Information 9 Relative abundance (%) of C. cumulans, M. strunckii, N corniger and T. riograndensis core microbiota.

Click here for additional data file.

PT, JDA, JC and MLM are CONICET members. The authors are grateful to Dr. Julia Sabio y García for linguistic improvement in the manuscript.

Additional Information and Declarations

Competing Interests

Author Contributions

Data Availability

Don A. Cowan was an Academic Editor for PeerJ.

Surendra Vikram conceived and designed the experiments, performed the experiments, analyzed the data, prepared figures and/or tables, authored or reviewed drafts of the paper, and approved the final draft.

Joel D. Arneodo conceived and designed the experiments, performed the experiments, analyzed the data, prepared figures and/or tables, authored or reviewed drafts of the paper, and approved the final draft.

Javier Calcagno conceived and designed the experiments, performed the experiments, analyzed the data, authored or reviewed drafts of the paper, and approved the final draft.

Maximiliano Ortiz conceived and designed the experiments, performed the experiments, analyzed the data, authored or reviewed drafts of the paper, and approved the final draft.

Maria Laura Mon conceived and designed the experiments, performed the experiments, analyzed the data, prepared figures and/or tables, authored or reviewed drafts of the paper, and approved the final draft.

Clara Etcheverry conceived and designed the experiments, performed the experiments, analyzed the data, authored or reviewed drafts of the paper, and approved the final draft.

Don A. Cowan conceived and designed the experiments, performed the experiments, analyzed the data, authored or reviewed drafts of the paper, and approved the final draft.

Paola Talia conceived and designed the experiments, performed the experiments, analyzed the data, prepared figures and/or tables, authored or reviewed drafts of the paper, and approved the final draft.

The following information was supplied regarding data availability:

Raw sequencing reads are available at the National Center for Biotechnology Information Short Read Archive: SRP152912 (16S rRNA) and SRP152912 (ITS region).

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
