# Peer review of "Diversity structure of the microbial communities in the guts of four neotropical termite species"

_PeerJ, doi:10.7717/peerj.10959_

## Round 0.1 · original submission · Major Revisions

Dear Dr. Talia,

Two reviewers were critical but positive about the potential of the manuscript, while a third reviewer rejects it. Please pay attention to all the suggestions of all reviewers and return a corrected version.

Reviewer 1 ·

Basic reporting

1. The authors should clearly state the aim of this study in the Introduction.

2. Fastq files of sequenced amplicons must be publicly available. Please provide the accession numbers.

3. L55: “lower” and “higher” termites is not defined based on their phylogenetic positions, not the presence/absence of flagellates.

4. Please rephrase L78 to “prokaryotic 16S rRNA gene and fungal ITS...” and remove the duplicated sentence “The analysis was based on ...” at L86–87.

5. L128, L188: Use lower case for “unweighted”.

6. L133–L135: I recommend describing the value of sequences and ASV counts in the Result section.

7. L175: Remove the parentheses.

8. L181 and elsewhere: “Eveness” and “Observed” should be rephrased to other terms such as “Pielou’s evenness” and “number of observed ASVs”, respectively.

9. Figure axis labels: SVSs -> ASVs

10. Because Fig. 3A and Fig. S1A are indicating substantially same things, Fig. 3A can be removed.

11. Table 2: Number of OTUs -> Number of ASVs

Experimental design

1. How did the authors identify the species of the termites?

2. L107–111: 515F and 806R primer pairs are for V4 region. Please cite the references of primers.

3. DADA2 have provided the official tutorial specific to ITS sequences (https://benjjneb.github.io/dada2/ITS_workflow.html). Did the authors conduct the ITS amplicon analysis in accordance with this tutorial? I am afraid that some artifact such as remaining primer sequences would affect the amount of the ASV-assigned reads. In addition, I ask the authors to show the read counts of raw fastq, quality-filtered fastq, and ASV-assigned reads in each sample.

4. L128: Which method was applied for the FDR correction?

5. L125–L128, L135–L139: It is confusing that methods for the community diversity calculations and statistic tests have been described in the two separate sections. Why did the authors apply two different approaches? Please provide a brief explanation.

Validity of the findings

1. Because as few as one or two termite species were examined for each diet group, the results shown in this study cannot exclude the possibility that not diet but host phylogeny is the major factor determining the structure of gut microbial community. In order to justify the authors’ argument “diet-driven microbial community assembly in the termite gut (L39)”, gut community of several more termite species should be simultaneously compared. Thus, I request the authors to compare the results from this study with those from previous researches, e.g., Mikaelyan et al., 2015; Rahman et al., 2015; Bourguignon et al., 2018. Especially, in addition to the discussion stated between L236–L256, I request a supplemental figure in order to prove that the results shown in this study are truly consistent with the previously published ones. I understand that it is often difficult to compare the results from different research projects due to the different methodologies, but it may be possible to preliminarily compare taxonomic compositions as shown in Fig. 1.

2. Given that each ASV represents a unique amplicon sequence and that the four examined termite species are affiliated with the distinct clades, it is quite surprising that 38 ASVs, representing as much as nearly 30% of total sequences, were shared among all of the four termite species. Rather, I suspect that these “core ASVs” are the results from DNA cross-contamination and/or cross-talk during sequencing (Wright and Vetsigian, 2016). To clarify the abundance of these core ASVs within each of 12 samples, I request the authors to provide a heatmap-style figure.

Additional comments

In this manuscript, Virkram et al. investigated the prokaryotic and fungal community structures in the gut of four higher termite species with different food preferences. 16S amplicon analysis revealed that prokaryotic taxonomic composition was distinct between soil-/grass- feeding termites and wood-feeding ones, and that species evenness and richness of prokaryotes were higher in the soil-/grass- feeding species than wood-feeding ones. Although most of fungal ITS amplicons were taxonomically unclassified, alpha-diversity indices suggest that fungal community is also distinct between soil/grass feeders and wood feeders. The authors concluded that diet is the major factor to determine the microbial community structures in the termite gut.
As referred in this manuscript, several studies had already investigated the relationship between gut microbial community, phylogeny, and diet of termites. These researches pointed out that both host termite phylogeny and diet can influence the community structures of gut microbiota. Thus, the discussion provided in this manuscript is not such novel; nevertheless this research may provide novel information on the gut microbial communities of not-yet-studied termite species, which would facilitate our deeper understanding on the ecology and evolution of termites and their gut symbionts. I would like to ask the authors to clearly explain the value and limitation of this study in the revised manuscript.

Mikaelyan et al. (2015) Mol. Ecol. https://doi.org/10.1111/mec.13376
Rahman et al. (2015) Microbiome https://doi.org/10.1186/s40168-015-0067-8
Wright and Vetsigian (2016) BMC Genomics https://doi.org/10.1186/s12864-016-3217-x
Bourguignon et al. (2018) Curr. Biol. https://doi.org/10.1016/j.cub.2018.01.035

Reviewer 2 ·

Basic reporting

The bacterial community structures in several higher termites including the investigated genera have already been studied elsewhere (such as Calusinska et al. (2020) and Marynowska et al. (2020) in addition to Dietrich et al. (2014) and Mikaelyan et al. (2015) in the references). These literatures should be considered in Introduction.

Calusinska et al. (2020) doi: 10.1038/s42003-020-1004-3
Marynowska et al. (2020) doi: 10.1186/s40168-020-00872-3

Experimental design

The compositions of the gut bacteria in the termites in this study were more or less different from the previous studies, while the authors assumed it partly due to variations in DNA extraction methods (L254-L255). The method described that the authors employed the DNeasy Blood and Tissue kit, but this kit does not contain reagents for lysis of the fungal and bacterial cell walls nor those for removal of humic compounds. The authors should clarify methodological descriptions of pretreatments to lyse the fungal and bacterial cell walls and remove humic compounds from soil/humus-feeders that inhibit PCR.

Validity of the findings

The conclusion from the results on the bacterial community structures was awkward because the present study analyzed the community structures with only four termite species, which belong to three different subfamilies and also have incongruous feeding preferences, making unlikely to be grouped to each other. In this context, I encourage the authors to analyze alpha- and beta-diversities shown in Fig, 3 with the bacterial community structures previously reported from several higher termites to obtain more a rigid conclusion.

Additional comments

This study analyzed the gut microbiomes of four Algentinian termite species and confirmed that diet determine the microbial community structure in termite gut. Their finding on the fungal phylum that was shared among the investigated termites seems novel and sound except for the procedure of DNA extraction mentioned above. However, the analyses on the bacterial community structures were somewhat awkward considering the current situation of this study area and should be improved taking the previous similar studies into consideration.

Reviewer 3 ·

Basic reporting

The manuscript describes a 16S rRNA/ITS based analysis of bacterial, archaeal and fungal gut communities associated with higher termites Cornitermes cumulans, Microcerotermes strunckii, Nasutitermes corniger, and Termes riograndensis. The primary hypothesis appears to have been to understand the role of diet in determining the gut community structure of these termite species, which has been addressed several times before, either using the exact termite species used in the current study or very closely related ones.

The manuscript is written clearly and I was able to follow the authors methodology. Most of the relevant literature has been cited, however, the authors do not cite it sufficiently in the introduction, which ends up overemphasizing the knowledge gap they are addressing. The article is clearly structured. Figures and tables have legible labels and captions. The results address the hypotheses adequately. However, as I remarked earlier, these hypotheses have been adequately tested in previous studies, which makes the results in the current study repetitive and unremarkable.

Experimental design

The research question:
The authors test the hypothesis that diet plays an important role in the determination of gut community structure in higher termites. This is a hypothesis that has been tested several times before with much larger datasets and using more sophisticated methods (1,2,3). While it may technically fulfill PeerJ's criterion for primary research, I would consider the study repetitive on account of selecting the exact same species used before and repeating a previously used methodology with a much smaller dataset.

Although the authors cite these and other studies as a token, they do not cite them in the right context in the introduction, which results in overemphasizing the knowledge gap. Moreover first-description statements as in 226-228 are highly inaccurate and misleading. N. corniger has the best assessed microbiomes among higher termites -not only do we know the presence and abundance of bacterial taxa in the hindgut (4), but in every gut section (5,6). It has been metagenomically (7) and metaproteomically (8) analyzed. We even know what bacteria are associated with wood fibers in the termite's hindgut (9). Moreover, C. cumulans has also been analyzed in great detail - taxonomically (10) and metagenomically (11.) We even know the microbiota associated with its mound (12). As for the remaining species, close relatives have been studied (1,3,6) in enough detail to give us an idea of at least their prokaryotic gut communities (which are by far the most ecologically relevant guild within the microbiome). In short, given all this published information, I am not sure a clear knowledge gap is being addressed.

However, despite the poor experimental design, the study is technically sound and has used state-of-the-art tools for processing sequencing data and statistical analysis. The methods described have been done so in enough detail that they can be replicated.


1. Mikaelyan, A., Dietrich, C., Köhler, T., Poulsen, M., Sillam‐Dussès, D. and Brune, A., 2015. Diet is the primary determinant of bacterial community structure in the guts of higher termites. Molecular Ecology, 24(20), pp.5284-5295.
2. Rahman, N.A., Parks, D.H., Willner, D.L., Engelbrektson, A.L., Goffredi, S.K., Warnecke, F., Scheffrahn, R.H. and Hugenholtz, P., 2015. A molecular survey of Australian and North American termite genera indicates that vertical inheritance is the primary force shaping termite gut microbiomes. Microbiome, 3(1), p.5.
3. Bourguignon, T., Lo, N., Dietrich, C., Šobotník, J., Sidek, S., Roisin, Y., Brune, A. and Evans, T.A., 2018. Rampant host switching shaped the termite gut microbiome. Current biology, 28(4), pp.649-654.
4. Dietrich, C., Köhler, T. and Brune, A., 2014. The cockroach origin of the termite gut microbiota: patterns in bacterial community structure reflect major evolutionary events. Applied and Environmental Microbiology, 80(7), pp.2261-2269.
5. Köhler, T., Dietrich, C., Scheffrahn, R.H. and Brune, A., 2012. High-resolution analysis of gut environment and bacterial microbiota reveals functional compartmentation of the gut in wood-feeding higher termites (Nasutitermes spp.). Applied and environmental microbiology, 78(13), pp.4691-4701.
6. Mikaelyan, A., Meuser, K. and Brune, A., 2017. Microenvironmental heterogeneity of gut compartments drives bacterial community structure in wood-and humus-feeding higher termites. FEMS Microbiology Ecology, 93(1).
7. Warnecke, F., Luginbühl, P., Ivanova, N., Ghassemian, M., Richardson, T.H., Stege, J.T., Cayouette, M., McHardy, A.C., Djordjevic, G., Aboushadi, N. and Sorek, R., 2007. Metagenomic and functional analysis of hindgut microbiota of a wood-feeding higher termite. Nature, 450(7169), pp.560-565.
8. Burnum, K.E., Callister, S.J., Nicora, C.D., Purvine, S.O., Hugenholtz, P., Warnecke, F., Scheffrahn, R.H., Smith, R.D. and Lipton, M.S., 2011. Proteome insights into the symbiotic relationship between a captive colony of Nasutitermes corniger and its hindgut microbiome. The ISME journal, 5(1), pp.161-164.
9. Mikaelyan, A., Strassert, J.F., Tokuda, G. and Brune, A., 2014. The fibre‐associated cellulolytic bacterial community in the hindgut of wood‐feeding higher termites (N asutitermes spp.). Environmental Microbiology, 16(9), pp.2711-2722.
10. Grieco, M.A.B., Cavalcante, J.J., Cardoso, A.M., Vieira, R.P., Machado, E.A., Clementino, M.M., Medeiros, M.N., Albano, R.M., Garcia, E.S., de Souza, W. and Constantino, R., 2013. Microbial community diversity in the gut of the South American termite Cornitermes cumulans (Isoptera: Termitidae). Microbial ecology, 65(1), pp.197-204.
11. Grieco, M.B., Lopes, F.A., Oliveira, L.S., Tschoeke, D.A., Popov, C.C., Thompson, C.C., Gonçalves, L.C., Constantino, R., Martins, O.B., Kruger, R.H. and de Souza, W., 2019. Metagenomic Analysis of the Whole Gut Microbiota in Brazilian Termitidae Termites Cornitermes cumulans, Cyrilliotermes strictinasus, Syntermes dirus, Nasutitermes jaraguae, Nasutitermes aquilinus, Grigiotermes bequaerti, and Orthognathotermes mirim. Current Microbiology, 76(6), pp.687-697.
12. Costa, P.S., Oliveira, P.L., Chartone-Souza, E. and Nascimento, A.M., 2013. Phylogenetic diversity of prokaryotes associated with the mandibulate nasute termite Cornitermes cumulans and its mound. Biology and fertility of soils, 49(5), pp.567-574.

Validity of the findings

Linking back to my comment in section 2 for an inadequately defined knowledge gap, I am not sure if the replicated sampling of the microbiomes of two well-studied species here is justified. Given that it has been shown that diet is the primary determinant of gut community structure in higher termites, I am not sure what contribution the current study makes (outside of some information on fungal communities) to literature.

The discussion is extremely rudimentary and superficial, and does not go into any detail about why they expect certain microbes to be dominant in a diet group. For instance, we know a lot about the microbial ecology of the guts of higher termites (1,2,3,4) and about how this may influence the gut community structure, but these valuable discussions have been left out.

1. Bignell, D.E. and Eggleton, P., 1995. On the elevated intestinal pH of higher termites (Isoptera: Termitidae). Insectes Sociaux, 42(1), pp.57-69.
2. Brune, A., Emerson, D. and Breznak, J.A., 1995. The termite gut microflora as an oxygen sink: microelectrode determination of oxygen and pH gradients in guts of lower and higher termites. Applied and Environmental Microbiology, 61(7), pp.2681-2687.
3. Köhler, T., Dietrich, C., Scheffrahn, R.H. and Brune, A., 2012. High-resolution analysis of gut environment and bacterial microbiota reveals functional compartmentation of the gut in wood-feeding higher termites (Nasutitermes spp.). Applied and environmental microbiology, 78(13), pp.4691-4701.
4. Mikaelyan, A., Strassert, J.F., Tokuda, G. and Brune, A., 2014. The fibre‐associated cellulolytic bacterial community in the hindgut of wood‐feeding higher termites (N asutitermes spp.). Environmental Microbiology, 16(9), pp.2711-2722.

---

## Round 0.2 · Minor Revisions

Please follow all the advice that reviewer provided, the manuscript is improving, but needs the last push.

Reviewer 1 ·

Basic reporting

No additional comment.

Experimental design

No additional comment.

Validity of the findings

No additional comment.

Additional comments

I appreciate the authors’ efforts to improve the manuscript, however there remains some points that should be addressed.

Fig. 5 BC and beta-diversity analysis
(i) Although the authors imply the influence of not only food habit but also host phylogeny on termite gut microbiome, the present manuscript lacks the analyses on the differences of gut microbiome in terms of host phylogeny. The authors should provide some statistical comparisons between host phylogeny (i.e., subfamily or genera) as did between the food habit.
(ii) It will be more convenient for readers if each sample point shown in the NMDS plots (Fig. 5BC) is categorized (shaped or colored) by host genera or subfamilies in addition to the food habit.
(iii) At least the result of unweighted UniFrac (Fig. 5B), I can find at least two clusters separated along the NMDS1 axis. Could the authors provide some interpretation of these clusters?

L140: Remove “V3”. 515F-806R primer pair is for the V4 region.

L181–L201: The methods described here are almost identical to what described in the section “Bioinformatic analysis of 16S rRNA and ITS sequences”. Please simplify them.

L196: due “to” the low number

L233: It is somewhat odd to mention Methanimicrococcus (Archaea) in the context of comparison between bacterial genera.

Fig.2 labels: “SVSs” have not been corrected to “ASVs” yet.

Fig. 6
(i) (A)Shared ... “four gut microbiomes” -> “four termite species”.
(ii) Remove “(B)Shared prokaryotic ASVs between four gut microbiomes”

Supplemental Figures: Numbering of supplemental figures is inconsistent with the main text.

---

## Round 0.3 · accepted · Accept

Thank you for this new version that is now ready for publication after addressing all the reviewer concerns.